# Detection of Interlayered Illite/Smectite Clay Minerals with XRD, SEM Analyses and Reflectance Spectroscopy

**DOI:** 10.3390/s22093602

**Published:** 2022-05-09

**Authors:** Fiorenza Deon, Frank van Ruitenbeek, Harald van der Werff, Mark van der Meijde, Camilla Marcatelli

**Affiliations:** Faculty of Geo-Information Science and Earth Observation (ITC), University of Twente, Hengelosestraat 99, P.O. Box 217, 7500 AE Enschede, The Netherlands; f.j.a.vanruitenbeek@utwente.nl (F.v.R.); h.m.a.vanderwerff@utwente.nl (H.v.d.W.); m.vandermeijde@utwente.nl (M.v.d.M.); c.marcatelli@utwente.nl (C.M.)

**Keywords:** interlayered clays, illite/smectite, X-ray diffraction, scanning electron microscopy, reflectance spectroscopy, gold epithermal deposits

## Abstract

Accurate determination of clay minerals can be challenging due to the natural occurrence of interlayered phases, i.e., layers of different clay species such as illite and smectite. The overlap of peaks of the constituent minerals (e.g., illite and smectite), and the similarity of diffraction patterns when not treated with ethylene glycol, hampers identification, especially when the clay content is low. We investigated the occurrence of interlayered illite/smectite in a rock sample from Rodalquilar, Spain, using X-ray diffraction, scanning electron microscopy and reflectance spectroscopy in the short-wave infrared wavelength range. For the first time, a precise determination of interlayered I/S conducted on the extracted clay fraction treated with ethylene glycol using such different approaches was provided. X-ray diffraction results demonstrated the presence of an I/S peak at around 8.4° in the untreated fraction coupled with a peak splitting at 6.7° and 9.4° 2θ when solvated with ethylene glycol. While spectroscopy indicated the occurrence of interlayered structures as a mixture of the two constituent minerals, the results of X-ray analysis showed that the interlayered clay consisted of two discrete phases (illite and smectite). The two discrete phases were observed in both the whole rock analysis and in the extracted clay fraction. This study shows that X-ray diffraction and validation with a scanning electron microscope is a mandatory, integrating tool for detecting interlayered phases since reflectance spectroscopy alone cannot be used to differentiate between interlayered clay minerals and non-interlayered mixtures. This work highlights the limits and advantages of three sensors (X-ray diffraction, scanning electron microscopy and reflectance spectroscopy) to investigate clay mixtures and interlayering, representing a significant contribution to confidence in the interpretation of interlayered clays, this being essential in mineral exploration and prospecting.

## 1. Introduction

Interstratified or interlayered illite-smectite (I/S) is a common sequence of clay layering in sedimentary rocks and mineralized areas. I/S are 2:1 dioctahedral phyllosilicates consisting of non-expandable illite layers and expandable smectite layers, as described by Wang et al. [1] and references therein. Those layers form an interstratification in different proportions, i.e., illite or smectite stacks prevail. The type of dominating layers, expandable (swelling-shrinking) and not-expandable, has a significant impact on geo-engineering infrastructure projects and also mineral-geothermal exploration [2,3]. Accurate detection of interlayering is challenging and time-consuming. Some problems may arise from peak overlapping in X-ray powder diffraction of I/S and palygorskite, a phyllosilicate, which share the first reflex at 8.4° 2θ. This makes the correct identification of an untreated sample quite problematic. This is why, among other factors, the application of ethylene glycol treatment is mandatory. Other difficulties are related to the amount of available clay in the samples and the time needed for extraction and sample preparation. In addition to mineral/geochemical/spectroscopic sensors, anomalies and occurrence of interlayering/swelling clays (water-bearing clays also in the subsurface) can also be detected with alternative approaches such as tomography of resistivity and wave velocity. The occurrence of clays or interlayered structures may result in anomalies in the subsurface which cannot be detected with traditional exploration methods [4]. Interesting results were recently presented by Dong et al. [4] where the application of three-dimensional (3D) tomography delivered a significant contribution to the research of subsurface structures and anomalies, improving the results of the available sensors. Considerable progress was achieved, with real-time monitoring coupled with conventional geophysical methods increasing the quality of subsurface imaging. Laboratory tests as described by Dong et al. [4] may also enrich the quality and sensitivity of sensors. The use of geophysical imaging is crucial for investigating the subsurface when geo-engineering projects and construction are involved [4]. The contribution of Ling Zeng et al. [5] also shows interesting results on the microstructural behavior of kaolinite-bearing mudstone in embankment engineering. An original mudstone was doped with cement, fly ash and red clay to investigate the response of the stress-strain parameters. The added clay component (red clay) induced no change in strain stress relationships. The work of Ling Zeng et al. [5] stresses the importance of the correct determination of clay-bearing rocks (i.e., carbonaceous mudstone), mineralogy and the investigation of associated microstructural properties applied to geo-engineering.

This work aims to deliver a practical contribution to the accurate identification of clays presenting interlayered structures. The application of X-ray diffraction (XRD), scanning electron microscopy (SEM) and reflectance spectroscopy leads to a complete and precise determination of the structures. These sensors deliver concrete help in the use of the exact clay as a geothermometer to gain information on the alteration pattern and hydrothermal alteration. The latter two factors play a crucial role in the constraint of mineralized ore areas. Clay minerals such as smectite, interlayered illite/smectite and illite are stable at different temperatures. Hence, the detection of interlayered structures delivers precious information for mining prospection. Reflectance spectroscopy in the short-wave infrared (SWIR) wavelength range can identify a mixture of illite and smectite, but cannot provide the exact contents of the two minerals in either interlayered or non-interlayered mixtures [2]. According to Inoue & Watanabe [6], the spectral mixing of illite and smectite in interlayered minerals is a non-linear process, which further complicates the unmixing of the spectra into the proportions of the constituent minerals. Simpson and Rae [2] have shown the limits of SWIR reflectance spectroscopy in the identification of interlayered-clay samples from the geothermal area of Taupo, New Zealand. Carillo-Rosúa et al. [7] reported the occurrence of interstratified clays in the Palai-Islica Au-Cu deposit, in the vicinity of Rodalquilar, South-east Spain. The results of this survey evidenced the occurrence of two different styles of alteration in the same hydrothermal system, both characterized by illite and smectite. This demonstrates how crucial the correct determination of interlayered clays and other phyllosilicates is when characterizing alterations in epithermal gold deposits, as they act as geothermometers. Investigation of the sources of illitization in I/S-bearing Pennsylvanian-aged paleosols [8] represents also a significant contribution to the understanding of I/S stability but also for paleoclimatic reconstructions. McIntosh et al. [8] investigate clay samples/fraction with one sensor using X-ray diffraction. A new thermodynamic calculation presented by Blanc et al. [9] on a natural illite/smectite series from a geothermal field in Shinzan, Japan, indicates an overall agreement with the previous mineralogical calculation. The application of enhanced reactive transport modelling enriches the available database, as in [9] and references therein. This enables the realistic forecast of the illitization processes via the formation of I/S with true reaction time in agreement with available mineralogical data. Recent study of clay-bearing rocks from Iran by Maghsoudi et al. [10] shows how different sensors, i.e., X-ray diffraction and reflectance spectroscopy, lead to different results from the same sample: while X-ray diffraction shows no evidence of smectite but only illite, reflectance spectroscopy detects both illite and smectite as discrete phases. Hence, the application of multiple sensors is often necessary to detect all rock mineral phases. Additional development in the characterization of clay minerals is thus needed to improve analyses of hydrothermally altered samples using multiple sensors.

To this date, few studies have been conducted comparing or combining the results of classical techniques with reflectance spectroscopy on samples from gold epithermal results bearing interstratified clays; hence, there are no records on any specimen from Rodalquilar where such a detailed approach was applied solely to identify interlayered structures. If, on one hand, reflectance spectroscopy coupled with X-ray diffraction on an extracted clay fraction might help to assess which clay forms the rock, then on the other hand there are considerable limits when it comes to distinguishing the interlayering and calculating the amount of illite and smectite in stacks. The novelty presented in this study is related to the correct application of the three presented methods and the limits of reflectance spectroscopy when investigating interlayered structures. As clays are used as geothermometers and help to localize mineralized areas, we have decided to test reflectance spectroscopy coupled with X-ray powder diffraction and scanning electron microscopy on fractionated clay. The selected sample for the clay extraction was collected from the epithermal gold deposit of Rodalquilar in Spain. The results of these sensors are compared and discussed in this work.

## 2. Methods

### 2.1. Provenance, Sample Description and Preparation

The Rodalquilar Caldera belongs to the Cabo de Gata volcanic field located in Andalusia, Spain (Figure 1). The prospect is well known for its high-sulfidation type gold epithermal deposits, intimately related to the caldera, as shown in Figure 1.

The occurrence of zeolite and an outcrop of bentonite rocks was described by Arribas et al. [13] and García-Romero et al. [14]. A description of the mineral assemblage of the Rodalquilar area was presented by Arribas et al. [13]. The surface rocks are characterized by quartz, k feldspar, illite, smectite, alunite, pyrophyllite and several Fe oxides such as hematite and goethite. Intense hydrothermal alteration led to the formation of typical mineral assemblages indicating intermediate to advanced argillite alteration. In particular, the intermediate argillic zone shows a halo characterized by the presence of a dominant illite-smectite horizon, as reported by van der Meer et al. [11] and references therein. Van der Meer et al. [6] also identified other key minerals in the Rodalquilar area, such as alunite, kaolinite, illite and chlorite by a combined use of airborne hyperspectral (HyMap) imagery and field-based ASD reflectance spectra.

In this study, one bright argillic altered volcanic rock bearing approximately 1 cm large sanidine crystals (Figure 2a), similar to the specimen described by García-Romero et al. [14], was investigated with X-ray diffraction for its smectite/illite content and potential interlayering of the two minerals. The sample was selected based on visible argillic alteration and previously reported composition of illite and smectite [15]. The rock showed intermediate to advanced argillic alteration and was classified as volcanic acidic rock type dacite to rhyolite; see García-Romero et al. [14] and references therein. In a first step, X-ray diffraction patterns were collected on whole-rock ground with an agate pestle mortar (Figure 2b).

One rock specimen (04MRE123) from the area, analyzed with X-ray diffraction (suitable for the clay extraction procedure and indicating a deviating value of the illite typical peak at 8.9° 2θ, see Section 3.1), was selected based on surface argillic alteration. The further analyses of its clay fraction were performed through X-ray powder diffraction, scanning electron microscopy and reflectance spectroscopy, analyzing for the presence of interlayered illite/smectite. In a second step, the selected sample was treated to extract clay phases using a pipette extraction procedure. The rock was first broken into small pieces, as shown in Figure 2b, and softened in water to avoid mechanical friction: no ball milling machine or pestle mortar was used, to preserve the integrity of the clay minerals. The rock was immersed in deionized water and heated up to 35 °C to ease the loosening of the finest particles in the water. The moisturized rock was wet-sieved; the separated fraction ≤63 µ was diluted with deionized water plus 1 g of a dispersive agent. The obtained 1 L solution was placed in a cylinder. The clay extraction was conducted following the pipette method described by Coates and Hulse [16]. After shaking and waiting for 8 h, the 2 µ suspension was withdrawn and placed in the oven to dry. The dispersive agent was rinsed after several drying procedures. The dried extracted fraction was weighted as 1.36 g and used for all analytical measurements, as described in Section 2.2, Section 2.3 and Section 2.4. The extracted material appeared as a white varnish coating, compact and challenging to remove from the holder. With gentle scratching, the dried fraction could be removed. The extracted clay fraction (≤2 µ) was air-dried and saturated with ethylene glycol at 60 °C overnight. The experimental procedure and scope of the analyses are summarized in the flowchart shown in Figure 3.

### 2.2. X-ray Diffraction (XRD)

The X-ray diffraction measurement on the whole-rock sample was conducted using an XRD Bruker D2 phaser (Bruker Corporation, Billerica, MA, USA) at the GeoScience Laboratory of the Faulty of Geo-Information Science and Earth Observation, University of Twente, the Netherlands. The equipment operated with CuKα radiation (1.54184 Å, 10 mA, 30 kV) and a LYNXEYE detector. The X-ray diffraction pattern was collected from 6 to 80° 2θ with 0.012° steps and an integration time of 0.1 s. In addition, a detector slit of 8 mm was used and a standard divergence slit of 0.6 mm was applied to control the illuminated area and enhance the resolution of the measurement. The phase identification and semi-quantitative calculation of mineral weight percentages were performed using the DIFFRAC.EVA software of Bruker. For the phase identification, the following structure files were used from the DIFFRAC.EVA library: sanidine (COD 900264); quartz (COD 1011097) and illite (COD 9013720).

The material chosen for the whole rock analysis was representative of the rock sample.

The X-ray diffraction measurements on the untreated and ethylene treated extracted clay fraction from sample 04MRE123 were collected with a Bruker D8 Advance at the University of Salzburg, Austria. The patterns were recorded with a step size of 0.012° respectively from 3 to 60° 2θ on the extracted fraction. The identification of interlayering is based on Reynolds [17].

### 2.3. Scanning Electron Microscope (SEM)

The extracted clay fraction was used for scanning electron microscope analysis. The sample was first spattered with carbon and placed on a sample holder. Backscattered electron (BSE) and secondary electron (SE) images were acquired with a ZEISS Ultra Plus Scanning Electron Microscope at the Helmholtz Centre Potsdam GFZ, Germany. Energy dispersive X-ray spectroscopy (EDS) spectra were collected with an acceleration voltage of 20 kV. The extrapolated chemical composition expressed in oxide % was normalized to 100%.

### 2.4. Reflectance Spectroscopy

Reflectance spectra were collected with a portable ASD spectrometer FieldSpec3 between 350 and 2500 nm wavelength at a spectral resolution of 7 nm at 700 nm and 10 nm at 1400 and 2100 nm. A white reference measurement was taken before each sample measurement. Two measurements were collected on the sample (whole-rock and extracted clay powder), which showed the occurrence of interlayered I/S determined via XRD on the clay fraction (see Section 3.2). Reflectance spectra of the following samples were acquired as well: extracted clay fraction (≤2 µ), whole rock and a well-characterized montmorillonite standard. The reflectance spectra were splice-corrected with software from ASD, and the SWIR portions of the spectra were further processed and analyzed with “The Spectral Geologist” software.

## 3. Results

### 3.1. X-ray Diffraction Whole-Rock

The X-ray diffraction pattern of the whole-rock sample 04MRE123 is shown in Figure 4a. The sample shows similar features with a dominant peak at 26.5° 2θ, which can be assigned to quartz, and a second sharp peak at 28°, assigned to sanidine. A bump can be observed at approximately 8.4° 2θ, as shown in the zoomed-in section in Figure 4b. Sample 04MRE123 was selected to investigate the occurrence of interstratified clays based on the detected peak at 8.4° 2θ, as illustrated in Figure 4b, and to conduct clay extraction. The mineral interpretation with the semi-quantitative abundances gave approximately quartz 40 wt.%, sanidine 40 wt.% and sheet silicates 20 wt.%.

### 3.2. X-ray Diffraction Clay Fraction

The pattern collected on the untreated (not solvated with ethylene glycol) and extracted fine fraction of sample 04MRE123 showed a sharp peak located at 8.4° 2θ (Figure 5a), reinforcing the results of the whole rock analysis. The value at this angle could neither be an indication of illite, usually at 8.9°, nor of smectite (or chlorite), usually located at 6.7° 2θ. Due to an enrichment of fine particles, the peak appeared higher and sharper than the other peaks that could be assigned to sanidine and quartz, as shown in Figure 5a. The presence of interlayered illite/smectite could be verified in the X-ray diffraction pattern of the ethylene glycol (EG)-treated clay fraction (Figure 5b). The pattern of the ethylene glycol fraction was characterized by peak splitting, with two peaks at 6.7° and 9.4° 2θ, related to the occurrence of illite/smectite (Figure 5b). The first peak at 6.7° 2θ was related to smectite (swelling) layers while the second peak at 9.4° was related to illite (non-swelling) layers.

Following this procedure, approximately 65% of the layers could be assigned to illite. The remaining 35% were assigned to smectite, indicating a dominant illitic component of no-swelling layers in the interlayering of illite and smectite. The abundances were calculated based on the shift of the peak at 8.4 to 9.4° 2θ after the ethylene glycol treatment of the irregular I/S layering as a function of the I/S ratio.

### 3.3. Scanned Electron Microscopy

Both backscattered electron images of the clay fraction of sample 04MRE123 show aggregates of small particles with different morphologies: Figure 6a indicates plate type aggregates formed by extremely small particles, whereas flakes aggregates indicate and validate the X-ray diffraction interpretation of the I/S occurrence. Smaller intergrowths with the same structure were already documented by Do Campo et al. [18] in backscattered images from ultra-polished thin sections, where they were assigned to interlayered structures. Figure 6b shows a more detailed morphology and size of the particles (2 µ) from the extracted sample fraction. Overall, the morphology was characterized by diffuse flakes up to 2 mm long, which were disposed on plates, respectively, from 1 up to 3 mm wide that we interpreted as interlayered I/S.

The energy dispersive X-ray spectroscopy spectra collected on two spots (Figure 7) indicated a chemical composition (oxide wt.%) dominated by approximately 65% SiO_2_, 30% Al_2_O_3_ and 5% K_2_O (for the atomic percentage, please see the Appendix A). However, spot 1 revealed an elevated K_2_O concentration (14 wt.%), thus indicating a composition closer to sanidine, as observed in the X-ray diffraction results. Chemical analyses on such small particles, especially clays, are often challenging due to the size of the beam diameter, which is often larger than the particles themselves. This may lead to the analysis of the immediate surroundings of the particle as well, thus obtaining a mixed analysis. Overall, SiO_2_ appeared as the most abundant oxide, also based on the spectrum intensity. Smaller concentrations (~1%) of FeO, CaO and Na_2_O were also detected. Due to the impossibility of measuring H_2_O via EDS, other elemental concentrations were required to differentiate I/S from sanidine at the micro-scale. Here we used the K_2_O abundances detected via scanning electron microscope to differentiate I/S from sanidine.

Additional measurements conducted with the EMP (electron microprobe) on the same ultra-polished thin sections of the samples provided similar results, indicating a predominant illitic composition of the analyzed specimen. The sparse occurrence of quartz and sanidine in the fine fraction could not be excluded based on the X-ray diffraction results.

### 3.4. Reflectance Spectroscopy

The reflectance spectra in Figure 8 were collected from (i) the intact rock sample 04MRE123, (ii) the extracted clay fraction of sample 04MRE123 and (iii) a pure well-characterized smectite (montmorillonite) compound for comparison. All three spectra show absorption features at approximately 1412 nm, 1907 nm and 2208 nm, which are induced by OH-stretching, interlayer molecular water and Al-O-H bending modes, respectively [19]. These features typically occur in both illite and smectite spectra. The spectra obtained from sample 04MRE123 (blue and red in Figure 8) also show another illite feature near 2343 nm [20], which is absent in the smectite spectrum (green in Figure 8). The whole rock reflectance spectrum (red in Figure 8) is characterized by relatively deep absorption features at 1412 and 2208 nm which are most similar to illite. The relatively deep water feature near 1907 could indicate a minor contribution of smectite as well.

The spectrum collected on the extracted clay fraction (blue spectrum in Figure 8) shows similar absorption features as the whole rock spectrum. However, the spectrum shows a generally decreasing slope of reflectance values at wavelengths longer than approximately 500 nm.

The smectite spectrum (green in Figure 8) differs from the spectra of rock sample 04MRE123 by the previously mentioned absence of the 2343 nm feature of illite and a relatively deep water absorption feature near 1907 nm compared to the Al-OH feature near 2208 nm.

Whether the reflectance spectra of rock sample 04MRE123 represent interlayered clay minerals or non-interlayered mixtures of illite and smectite cannot be determined from these spectra alone.

## 4. Discussion: Interlayered Clays Detection with X-ray Diffraction, Scanning Electron Microscopy, Reflectance Spectroscopy or All?

Precise identification of clays and interlayering is important in mineral prospection because it helps to localize alteration halos and determine the position of ore minerals. White and Hedenquist [3] describe how the spatial variation of clay mineral composition is, among others, the best factor for calculating paleotemperature, a crucial indicator for mineral prospects. Smectite, which is stable at 160 °C, gradually transforms into interlayered illite/smectite with increasing temperature. A further temperature increase (>220 °C) changes interlayered I/S into illite [21,22]. A complete assessment of the alteration mineralogy of gold epithermal deposits is therefore crucial, as already stated by White and Hedenquist [3]. The temperature of stability of interlayered I/S corresponds to the temperature of the epithermal ore deposition (150–300 °C), as described by White and Hedenquist [3]. It appears how intimate the interlayered clay occurrence is with the formation of gold deposits and highlights the importance of precise detection of an alteration pattern typical of the deposit.

Traditional mineral analyses techniques are microscopy of thin sections, X-ray diffraction and backscatter techniques. The application of reflectance spectroscopy has increased in the last decades [23] as it allows fast mineral identification in the field and laboratory.

Nevertheless, reflectance spectroscopy and remote sensing techniques in general also present limitations when it comes to the identification of interlayered structures, i.e., I/S, as described below, and swelling clays, i.e., montmorillonite [24].

The example provided by Maghsoudi et al. [10] clearly shows how different sensors may provide different results from the same sample containing clay minerals (illite and smectite). Factors influencing results could be the detection limit of the individual techniques used (X-ray diffraction ~ 2–3 wt.%), the sample preparation, the degree of alteration or the occurrence of amorphous phases.

The authors paid particular attention to the sample preparation by avoiding any possible factor that might affect the clays’ crystallinity, i.e., ball milling machine. As shown in Section 2.1, the portion of 04MRE123 selected for clay extraction was neither milled nor ground with pestle and mortar. Mechanical procedures applied on clays may influence results by changing or destroying the layers of illite and smectite and, consequently, any measured X-ray diffraction pattern. Maghsoudi et al. [10] present a significant difference in their output: X-ray diffraction shows no montmorillonite but only illite while reflectance spectroscopy both illite and montmorillonite. Our work instead highlights how two sensors, X-ray diffraction and scanning electron microscopy, detect the interlayered I/S, while reflectance spectroscopy can only show the presence of both discrete phases, thus missing important results when interlayered clays may be responsible for an anomaly in the subsurface. At a major scale (i.e., not one sample), the application of different sensors such as tomography is therefore recommended to upscale results and detect anomalies (swelling clays horizons or discontinuities) as described by Dong et al. [4].

This work shows, for the first time, how combined measurements by X-ray diffraction, scanning electron microscopy and reflectance spectroscopy can be used to analyze an extracted clay fraction for interlayered clays. Results indicate that reflectance spectra (Figure 8) cannot distinguish whether clays in a sample occur as a non-interlayered mixture or as single interlayered clay minerals. The use of solely reflectance spectroscopy can work if one of both phases occurs in a hydrothermally altered sample but might lead to a misinterpretation of the clays, characterizing the alteration pattern of the epithermal gold deposit if interlayered structures occur. Even though the reflectance spectrum shows dominance of illite, the rock itself is composed predominantly of sanidine and quartz (see Figure 4a). These minerals are not active in the SWIR wavelength range and therefore cannot be observed with reflectance spectroscopy in the VNIR-SWIR wavelength range.

The authors are aware that, in mineral prospection, it is economically impossible to systematically apply all these methods. However, this should be implemented whenever surface rocks indicate clay mineral abundances and spectra indicate a mixture of illite and smectite.

The spectra in Figure 8 were compared with X-ray diffraction to a well-characterized smectite standard: observing the reflectance spectra reveals that there are slight differences in the presence and (relative) depths of absorption features. The X-ray diffraction pattern on the smectite standard indicates a characteristic peak at 6.7° 2θ and a shift toward lower 2θ numbers typical for smectite. The X-ray diffraction pattern of 04MRE123 shows a clear difference in peak position and behavior after ethylene glycol treatment, indicating interlayering. The differences in the reflectance spectra are not sufficiently strong to distinguish interlayering from pure illite or smectite.

The reflectance spectroscopy spectrum collected from the 04MRE123 clay fraction shows an atypical slope. The measurement was collected from extremely fine material (2 µ) and the grain size might have affected the related spectrum. Clark and Roush [25] and references therein report interesting phenomena related to the very small particle size (1 µ) of very fine frost, where a linearly decreasing trend with the wavelength characterizes the reflectance spectrum. An obvious question rise: how to correctly identify interlayered I/S? Although only a single sample was investigated with the three analytical methods, this study provides the first attempt to deliver a practical answer to the question by showing that X-ray diffraction combined with scanning electron microscopy can be used to identify interlayered illite-smectites, which reflectance spectroscopy may not be able to do. The use of all three methods provides complete detection of all mineral abundances, including interlayered clays.

## 5. Conclusions

This study delivers a small but significant contribution to improving detailed mineralogical investigations to be used in mineral prospection strategy and planning for gold epithermal deposits, where the identification of illite-smectite interlayering plays a critical role.

Reflectance spectroscopy works on pure mineral phases but shows limitations when interlayering occurs. Spectroscopy cannot convincingly indicate the occurrence of interstratification but only indicate the presence of discrete phases or a mineral mixture. Hence, analysis through X-ray diffraction and scanning electron microscopy should always be performed when working with clay minerals to complete spectroscopic interpretation.

## Figures and Tables

**Figure 1 sensors-22-03602-f001:**
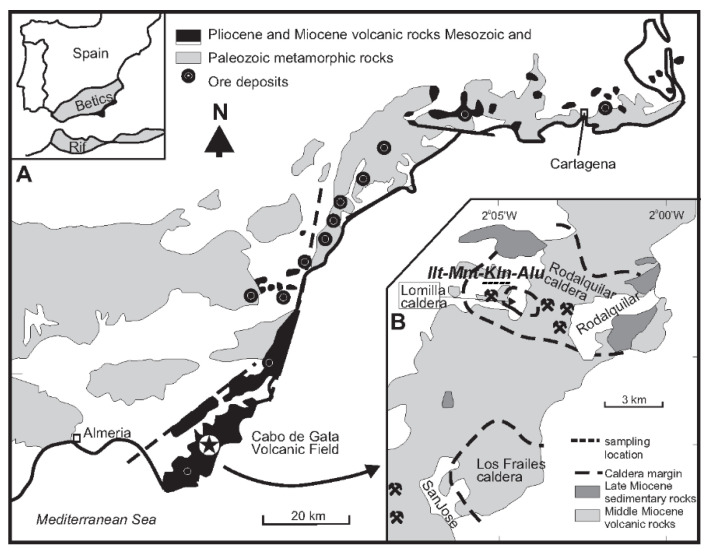
(**A**) The Rodalquilar area with sample locations. (**B**) Detail of the area with typical alteration minerals for epithermal gold deposits: illite (Ilt), montmorillonite (Mnt), kaolinite (Kln) and alunite (Alu). Modified with permission after van der Meer et al. [11]. Mineral abbreviation according to Whitney and Evans [12].

**Figure 2 sensors-22-03602-f002:**
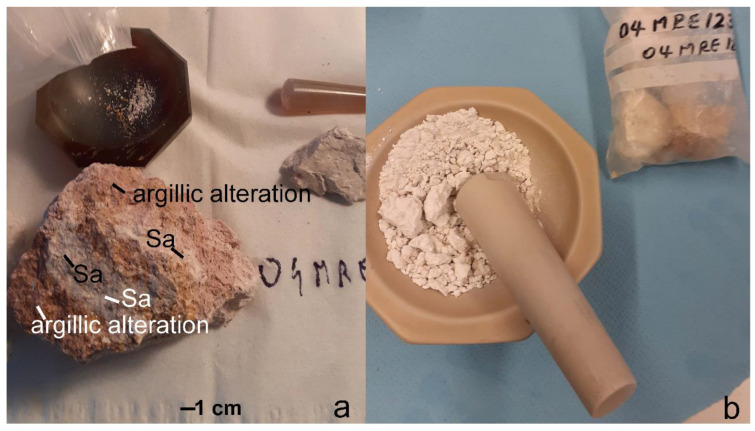
(**a**) Rock specimen 04MRE123, selected for clay extraction, where the occurrence of sanidine and argillic alteration is shown. (**b**) Hammered rock pieces in the agate mortar before X-ray diffraction, whole-rock measurement and clay extraction procedure.

**Figure 3 sensors-22-03602-f003:**
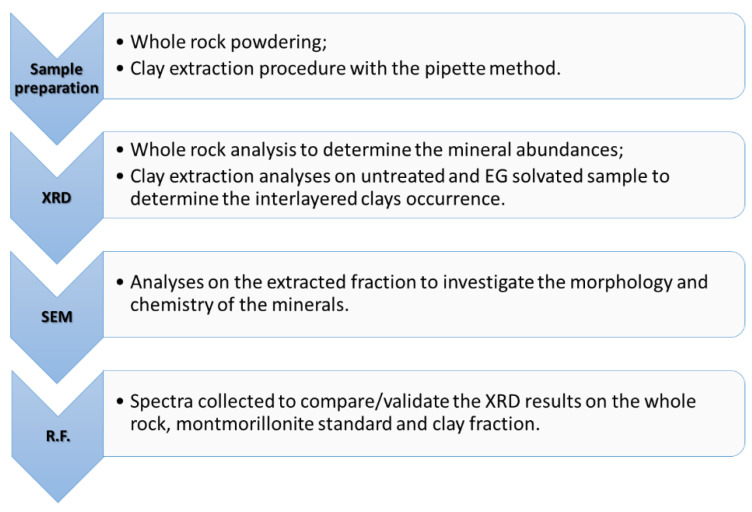
Flow chart indicating the workflow (R.F. = reflectance spectroscopy).

**Figure 4 sensors-22-03602-f004:**
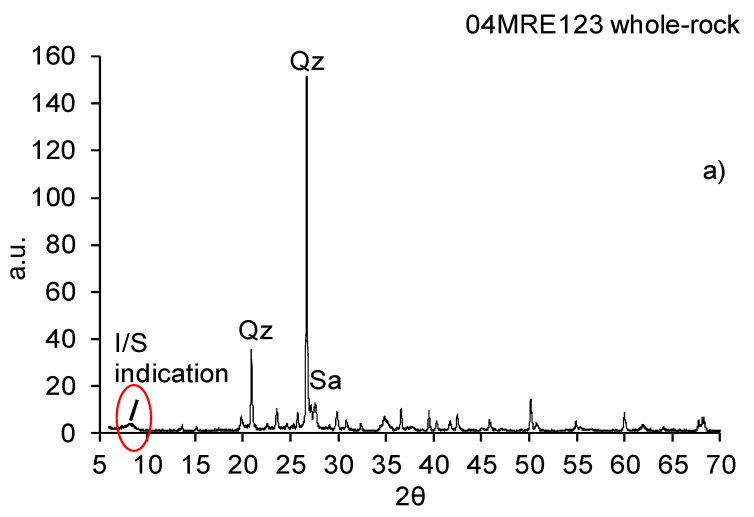
(**a**) Whole-rock X-ray diffraction pattern of the sample 04MRE123 from Rodalquilar. The pattern shows a sharp quartz peak at 26.5 °2θ and a sanidine peak at 28° 2θ. The peak at 8.4° 2θ indicate the interlayering of illite and smectite. (**b**) Zoomed-in section of the whole-rock X-ray diffraction pattern where the peak position of the I/S indication is shown.

**Figure 5 sensors-22-03602-f005:**
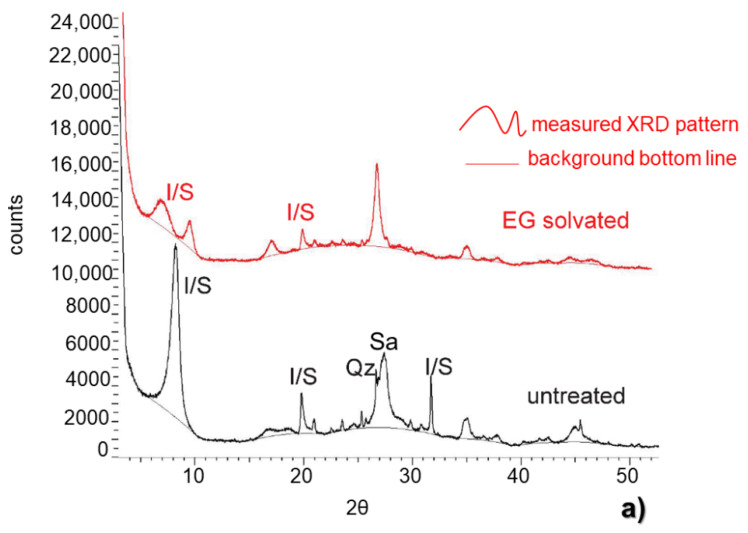
(**a**) X-ray diffraction patterns of the untreated and treated extracted fraction of sample 04MRE123, with the I/S peak signature at 8.4° 2θ. Additional peaks were assigned to sanidine and quartz. (**b**) The enrichment in the clay component is visible in the intensity of the peak at 8.4° 2θ, compared to the height in the whole rock pattern of the same samples, as shown in Figure 4a,b). The pattern collected after glycol treatment shows the occurrence of two peaks representing a split of the reflex at 8.4° 2θ: the peak at 6.7° 2θ is assigned to the smectite component of the interlayering while the second peak located at 9.4° 2θ is assigned to the illite fraction of the interlayering.

**Figure 6 sensors-22-03602-f006:**
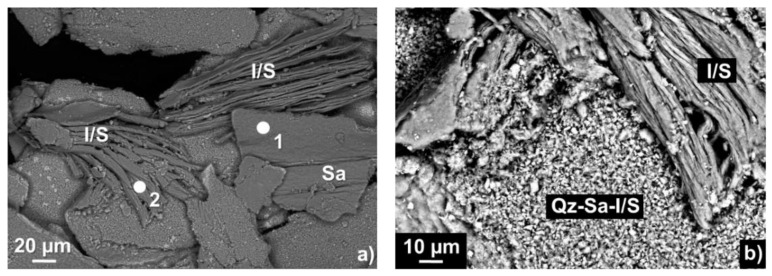
(**a**) Backscattered electron image of the clay fraction of sample 04MRE123 with sanidine (Sa) and elongated illite/smectite (I/S) flakes. The white dots (1 and 2) indicate spots where energy dispersive X-ray spectroscopy spectra were collected. (**b**) Backscattered electron image showing fine aggregates of sanidine, quartz and I/S (center of the image) as well as large I/S flakes.

**Figure 7 sensors-22-03602-f007:**
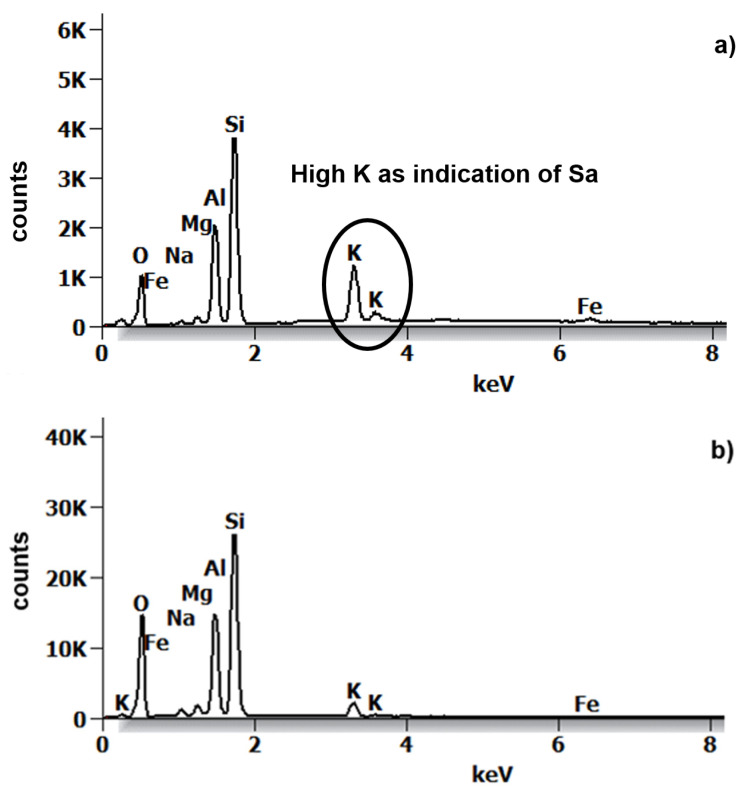
Two Energy dispersive X-ray spectroscopy spectra acquired from sample 04MRE123 extracted fraction. Analyses correspond to the position 1 and 2 shown in Figure 6a. (**a**) The spectrum 1 indicates sanidine based on the peak ratios and a higher K peak. (**b**) The spectrum can be assigned to a clay phase (very low K intensity), in our case interlayered I/S based on the mineralogy, morphology and chemistry.

**Figure 8 sensors-22-03602-f008:**
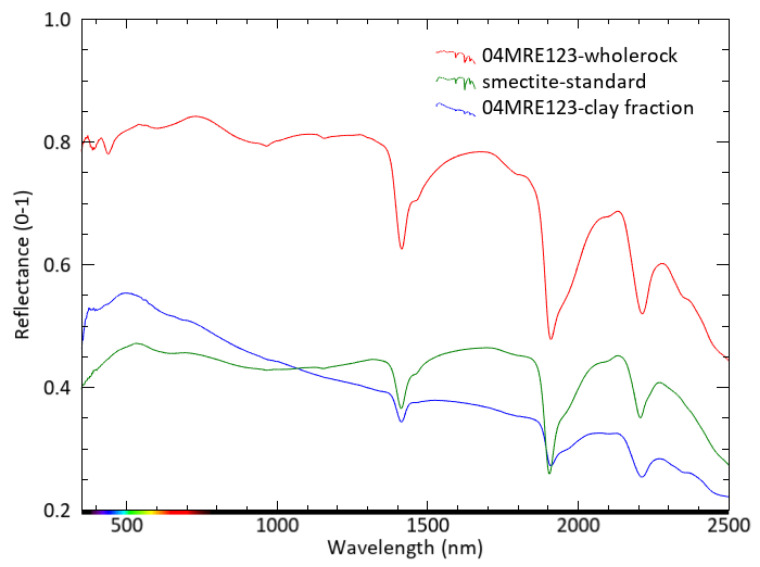
Reflectance spectra of the intact whole-rock sample 04MRE123, its clay fraction and a smectite standard.

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
