# Peer review of "Detection of Interlayered Illite/Smectite Clay Minerals with XRD, SEM Analyses and Reflectance Spectroscopy"

_sensors, 2022, doi:10.3390/s22093602_

Round 1

Reviewer 1 Report

See the attached file including general and specific comments.

Reviewer 2 Report

Review for

sensors-1668225

Detection of interlayered illite/smectite clay minerals with XRD, SEM analyses and reflectance spectroscopy

Accurate determination of clay minerals is important. The method presented in this work is exciting and could be significant in practical application. I think, nonetheless, that the manuscript could be improved if the authors could address the comments and recommendations I listed below.

The novelty of this research should be highlighted in the Abstract.

Line 35: Wang and Wang should be rewritten in Wang et al. 

In the intro part, please describe the detailed information about why accurate detection of interlayering is challenging? What are the typical problems?

Please highlight what is the connection between senors and your research. Why do you want to publish your article in the senors journal?

Line 103. I recommend you add a scale bar in figure 2a.

In figure 4. It is very hard to read your XRD spectrum. Will you be able to use an A.U. y-axis to overlay your spectrum (like your figure 5)? Additionally, you should have a citation or standard powder diffraction file(PDF) for your XRD  peaks. 

In figure 6. some typos in your Fig 6 c,d. Additionally, you did not mark a,b,c,d in your figure.

Do you have atomic percentage data in your EDS result?

Finally, your study is interesting, but all your figures are presented in low quality. Please revise it.

Reviewer 3 Report

The work investigated the occurrence of interlayered illite/smectite in a rock sample from Rodalquilar, Spain, using X-ray diffraction, scanning electron microscopy and reflectance spectroscopy in the Short Wave Infrared wavelength range. It is a carefully done study and the findings are of considerable interest. However, I thought it still has some deficiencies and I recommend to a revision before acceptable publication. Detailed comments are listed below:

--Section 1: A more detailed description of previous research in introduction is needed, especially the published literatures on identify interlayered structures in other method, for example: Quantitative investigation of tomographic effects in abnormal regions of complex structures. Engineering, 2021, 7(7): 1011-1022. Mechanical behavior and microstructural mechanism of improved disintegrated carbonaceous mudstone. Journal of Central South University, 2020, 27(7): 1992-2002.

--Section 1:The literature on detection of interlayered illite/smectite clay minerals is not summarized enough, and the innovation of the paper is not reflected.

--Section 2: Compared with other mineral identification methods, what are the advantages of XRD, SEM analyses and reflectance spectroscopy chosen in this paper? What are the differences from them?

--Section 2: Are the materials selected in the text representative? Does its size make a difference to the measurement? Please make it clearly.

--Section 2: Is it too cumbersome to use only words in the process of analyzing the selected samples? Have you considered using a flow chart model to represent the processing of the sample?

--Section 2: The XRD test often reflects the macroscopic properties of the sample, but is it reasonable to judge only by the macroscopic appearance? In addition, too few samples may also affect the analysis of the results. How to deal with such a problem? Please make it clearly.

--Section 2: How to prevent distortion of measurement results when analyzing clay composition with Reflectance spectroscopy? How to ensure the accuracy of the recognition results?

--Section 3: Since different methods have different starting angles, how to compare the advantages and disadvantages of each method?

--Section 3: Are there settings for observations when analyzing clay composition using different methods? What standard is used to determine the clay composition?

Round 2

Reviewer 2 Report

After checking the draft of the response to the comments, and the corresponding revisions in the revised manuscript, I found that the authors have accomplished the recommended revision to address all my concerns.

This article is good to go.

Reviewer 3 Report

The work investigated the occurrence of interlayered illite/smectite in a rock sample from Rodalquilar, Spain, using X-ray diffraction, scanning electron microscopy and reflectance spectroscopy in the Short Wave Infrared wavelength range. I disagree some of the authors reply. For example, as a paper for possible publication, some related works/methods on your topic should be summarized and discussed. For your paper, the tomography of resistivity and wave velocity can also be used to reach your study goals. So I think you should include these methods to make up for the lack of sensor-related content. There are also too few references in the last five years, Please revise. Otherwise, I think this work is not suitable for publication in sensor-related journal.
